# Sputtering Plasma Effect on Zinc Oxide Thin Films Produced on Photopolymer Substrates

**DOI:** 10.3390/polym15102283

**Published:** 2023-05-12

**Authors:** Juan Jesus Rocha-Cuervo, Esmeralda Uribe-Lam, Cecilia Daniela Treviño-Quintanilla, Dulce Viridiana Melo-Maximo

**Affiliations:** 1Tecnologico de Monterrey, School of Engineering and Sciences, Estado de Mexico Campus, Atizapán 52926, Mexico; a01752555@tec.mx (J.J.R.-C.); virimelo@tec.mx (D.V.M.-M.); 2Tecnologico de Monterrey, School of Engineering and Sciences, Queretaro 76130, Mexico; cdtrevino@tec.mx; 3Tecnologico de Monterrey, Institute of Advanced Materials for Sustainable Manufacturing, Ave. Eugenio Garza Sada 2501, Monterrey 64849, Mexico

**Keywords:** plasma, UV treatment, sputtering, zinc oxide (ZnO), thin films, stereolithography (SLA), photopolymerization

## Abstract

This work presents a post-cured treatment alternative for photopolymer substrates considering the plasma produced via the sputtering process. The sputtering plasma effect was discussed, analyzing the properties of zinc/zinc oxide (Zn/ZnO) thin films deposited on photopolymer substrates, with and without ultraviolet (UV) treatment as a post-treatment process, after manufacturing. The polymer substrates were produced from a standard Industrial Blend resin and manufactured using stereolithography (SLA) technology. After that, the UV treatment followed the manufacturer’s instructions. The influence of the sputtering plasma as an extra treatment during the deposition of the films was analyzed. Characterization was performed to determine the microstructural and adhesion properties of the films. The results showed the effect of plasma as a post-cured treatment alternative: fractures were found in thin films deposited on polymers with previous UV treatment. In the same way, the films showed a repetitive printing pattern due to the phenomenon of polymer shrinkage caused by the sputtering plasma. The plasma treatment also showed an effect on the thicknesses and roughness values of the films. Finally, according to VDI–3198 standards, coatings with acceptable adhesion failures were found. The results provide attractive properties of Zn/ZnO coatings on polymeric substrates produced by additive manufacturing.

## 1. Introduction

Surface engineering produces thin films to improve one or several surface properties of solid materials [1]. Chemical vapor deposition (CVD), Physical vapor deposition (PVD), and sol–gel are examples of technologies used to produce thin films. In addition, several techniques for thin film production are based on PVD [2,3,4], such as Reactive magnetron sputtering (PVD-RMS). PVD-RMS allows the production of high-density coatings (e.g., metallic, ceramic, etc.) with excellent mechanical properties on several substrates (e.g., polymeric, metallic, ceramic, etc.) to provide unique surface properties in materials for various engineering applications [5,6].

PVD-RMS is an assisted plasma technology that requires the formation of a plasma environment, in which the sputtered particles will be transported to produce a thin film on a substrate. The plasma in the PVD-RMS technique is composed of ionized gas, pulverized particles of the target in constant collision, secondary electrons, UV light, visible light, and heat [7,8,9,10,11].

Some research discussed the influence of the substrate surface finish and the topology on the thin film growth pattern. In coating growth phenomena, the particles of the film adhere and adapt to the substrate surface where they are deposited, mimicking the substrate’s surface topology [12,13,14]. Types of substrate materials used in PVD-RMS include metals, glass, polymers, and composites. In the case of polymeric materials, they can be used as substrates for the deposition of thin films regardless of their production method. For example, polymers developed in additive manufacturing (AM) by three-dimensional (3D) printing techniques have been used in different types of analyses to improve surface properties by adding a thin film [15,16,17].

Polymers could be processed via several AM techniques, such as stereolithography (SLA), digital light processing (DLP), and continuous liquid interface production (CLIP). The SLA technique is based on the photopolymerization principle to develop 3D designs into prototypes and functional objects [18]. A UV light source, usually a laser, initiates a chain reaction on a photocurable liquid resin. The laser polymerizes and solidifies one layer at a time until the desired solid part is obtained [19]. The initial polymerization process via the printer laser provides solid specimens with an excellent aesthetic finish, fine resolution (as low as 5 μm), and adequate handling strength that can be directly used for different applications [20]. The patterns produced on the final sample via SLA printers, presented in Figure 1a, are not fully polymerized from exposure to UV light in the printing process, resulting in a small volume of unpolymerized resin throughout the matrix of the pattern [21].

The pattern is known as the “green-state structure” and is characterized as being a material with different curing portions throughout the entire structure. Since the final piece is not fully polymerized, a post-cure method is often recommended to improve the final properties of the AM part [20,21,22,23,24].

Research analysis and experimentation determine that the plasma environment in sputtering techniques can reach temperatures between 533 and 813 K [7,9,10,11,25,26,27]. The heat produced via temperature variation can transform the surface of the polymeric substrate used for thin film production [24,28]. Considering that photosensitive polymers are susceptible to changes in properties due to high temperatures, it is important to consider these materials’ limitations when used as substrates to produce thin films via plasma-assisted technologies [29]. Therefore, photosensitive resins receive a post-cure process that can be replaced during the PVD-RMS thin film production due to the nature of plasma.

This article aims to analyze the effect of plasma produced via the PVD-RMS technique, as an alternative post-curing treatment compared to the conventional UV treatment, on the microstructural and adhesion properties of a Zn/ZnO thin film on a photosensitive polymer substrate. The produced thin films were characterized by optical microscopy (OM), scanning electron microscopy (SEM) equipped with energy dispersive x-ray spectroscopy (EDS), and atomic force microscopy (AFM). The coating substrate’s adhesion was measured using a Rockwell durometer, following the VDI 3198 standard.

## 2. Materials and Methods

Cylindrical-shaped polymeric substrates (25 × 5 mm) were produced using an acrylic and glycol diacrylate-based photopolymer resin (Industrial Blend^®^ developed by FunToDo^®^) on a stereolithography printer (Elegoo Mars UV Photocuring printer LCD MSLA, Shenzhen, China). The summary of the printed parameters is in Table 1.

The printed substrates were cleaned with an ultrasonic bath (Ultrasonic Cleaner Model JPS-10A, Shenzhen, Guangdong, China) in isopropanol (99.999%) for 20 min and dried with pressurized air. A UV treatment using UV light (90–260 V UV lamp, 405 nm, 60 W light effect, Shenzhen Enomaker Technology Co., Ltd.; Longhua Dist, Shenzhen, China) for 10 min was performed on substrate 1 as the manufacturer recommended [30], and substrate 2 was directly used after the dried stage in the printing process.

Zn/ZnO coatings were deposited on substrates 1 and 2 via Physical Vapor Deposition with unbalanced Reactive Magnetron Sputtering technology (PVD-RMS) in Ar and O_2_ mixed atmosphere. The deposition equipment and the principal components are presented in Figure 2; details of the system have been described elsewhere [31].

Circular zinc (Zn) plate with a purity of 99.99% was selected as a target [2″ (5.08 cm) diameter × 0.25″ (0.635 cm) thick]. The polymeric substrates were placed into the reactor at ~30 mm distance from the target. After placing the substrates, the sputtering chamber was evacuated to a base pressure of ~1 × 10^−4^ Torr (1.3 × 10^−2^ Pa). Argon was introduced into the chamber at a constant flow rate of 20 SCCM (standard cubic centimeter per minute), leading to an increase in pressure around 4 Pa. In these conditions, a power of 40 W was then applied to the magnetron to initiate the process, maintaining the substrate at room temperature. The target was cleaned in these conditions for 5 min, and the pressure was adjusted to 2 Pa. A pure Zn adhesion interlayer was deposited at 40 W for 1 min in a pure Ar atmosphere. The power was reduced to 30 W to produce the ZnO layer, and the oxygen was introduced into the chamber at a constant flowrate of 7.3 SCCM. The ZnO layer was deposited for 5 min in an argon and O_2_ mixed atmosphere. The total deposition time was 6 min for all the samples.

Optical microscopy (OM), Olympus Corporation, Shinjuku Tokyo, Japan, was used for the superficial inspection of samples (PGM 3 Olympus microscope) for characterization analysis. The morphology examination of the Zn/ZnO films was performed via scanning electron microscopy (SEM JEOL JSM-6360 LV) Jeol, Ltd. Akishima, Tokio, Japan, equipped with energy-dispersive spectroscopy analysis (EDS) Oxford Instruments, Santa Barbara, CA, USA, for coating chemical composition evaluation. Surface topography was characterized via atomic force microscopy (AFM—Park Systems model XE7) Park Systems Corp, Suwon-si, Gyeonggido, Korea, operating in Non-Contact Tapping mode, using a Si probe, and scanning areas of 20 × 20 and 5 × 5 μm^2^. Adhesion of the films was evaluated by the VDI 3198 standard, using a Rockwell durometer (Louis Small model 8SSA) Louis Small Inc., Cincinnati, OH, USA; a load of 150 kg was applied with a diamond tip in the C scale.

## 3. Results and Discussion

### 3.1. Effect of UV Treatment on the Surface Properties of Zn/ZnO Thin Films Deposited on Photosensitive Polymeric Substrates

The samples were examined after the Zn/ZnO thin film deposition to analyze the effect of the PVD-RMS plasma on the surface properties of the films. Polymeric substrates were used, with 10 min of UV treatment post-process after manufacturing [30] and as manufactured without UV treatment, as mentioned in Section 2, “Materials and Methods”. Subsequently, the substrates were subjected to a PVD-RMS process assisted with plasma to produce the Zn/ZnO thin film.

Figure 3 presents the OM images, at high magnification, of the ZnO thin films deposited on the polymeric substrates used, showing the characteristic surface topology produced via the SLA printer on the substrate surface of the samples. The results presented a difference between the analyzed thin films; the thin film deposited on the UV-treated polymeric substrate presents fracture cracks along the surface of the sample. In contrast, the thin film deposited on the not treated polymeric substrate presents a pattern without changes after deposition.

After applying 10 min of UV treatment, as mentioned in Section 2, “Materials and Methods” [30], fractures were presented along the sample’s surface. The fracture cracks shown in Figure 3a were related to a deterioration of the polymeric substrates due to an abrupt change in temperature of the samples by direct contact with the PVD-RMS plasma environment, also associated with over-curing phenomena [21,32,33,34,35,36,37,38]. The ZnO thin film deposited on the photosensitive polymeric substrate without UV treatment is shown in Figure 3b. Due to growth phenomena, the thin film adheres and adapts to the substrate surface without modifying the characteristic printing pattern [12,13,14]. In the same way, the thin film looks homogeneous in all sections without the presence of apparent fractures.

Post-cure treatments can increase the mechanical properties of photosensitive polymers via polymerizing uncured resin portions trapped within the patterns and promoting the polymerization of the already cured in the ”green-state structure”, thus increasing the crosslinking density [23,39]. Nevertheless, restrictions may apply when UV treatments are replaced by heat treatments concerning other polymer properties such as heat deflection resistance, temperature resistance, and thermal shock resistance [21,29]. Several studies [29,40,41] have demonstrated that long exposure UV treatments do not produce remarkable changes in the final mechanical properties of photosensitive polymers, i.e., compressive strength, tensile strength, and Young’s modulus reach a limit after several hours of UV treatment. However, in the case of heat treatments, if temperatures are raised above the thermal limit of the photosensitive polymers, or the photosensitive polymers are exposed to abrupt changes in temperature, fractures may appear [21]. Various studies [28,29,42] have shown that when green-state pieces are cured at high temperatures for short times, shrinkage produces fractures that weaken the final and surface properties of the desired polymer part. Therefore, it becomes necessary to increase the temperature of the post-cure treatment based on the limitations of the polymer used [22,29].

As the plasma in the PVD-RMS technique is composed mainly of ionized gas, the free electrons in its constitution move at very high velocities because of lower mass and high temperature. These energetic electrons can deliver significant power, either in the form of UV radiation or temperature, thus resulting in substrate heating during the deposition process [43,44].

According to the manufacturer, Industrial Blend^®^ resin polymers are characterized as withstanding temperatures up to 498 K [30]. However, as some studies have shown [21,25,29,30], the plasmas in the sputtering technique can reach temperatures between 533 and 813 K. Therefore, it is suggested that the plasma–substrate interaction produced a thin film with degraded surface quality, as shown in Figure 3a, generated by changes in substrate temperature upon contact with the PVD-RMS plasma, leading to high residual stresses and originating the observed fractures.

Invariably PVD-RMS films have residual stresses that arise from the growth processes [45]. Nevertheless, it has been registered that differences in the thermal coefficients of expansion of the film and substrate in high-temperature depositions may cause plastic deformation, cracking the thin film or the substrate, or cracking at the substrate–film interface [10,45]. Although in this research thin films were not deposited at high temperatures, the energy and heat generated by the nature of the PVD-RMS plasma inevitably produced changes in the polymeric substrate, inducing constant modifications on the thin film during its deposition, allowing the high accumulation of stresses, and originating the observed fractures.

Microstructural characterization and surface analysis of as-deposited Zn/ZnO thin films were conducted via SEM technology to determine how the PVD-RMS plasma conditions change the coating topography of Zn/ZnO thin films deposited on both; a photosensitive polymeric substrate with 10 min of UV treatment before the deposition of the thin film, and a polymeric substrate directly used without UV treatment. Figure 4a,b presents backscattered electron (BSE) SEM micrographs, at high magnification, to compare the main differences found in topography. Regardless of changes in UV treatment, dense, compact, and well-adhered Zn/ZnO films were obtained; neither fractures nor detachment was found in the thin film in the analysis.

As observed from SEM images, it was determined that the topology of the films changed depending on the substrate treatment. From Figure 4a, a homogeneous growth in the form of a labyrinth is observed, presenting a lower density since not only a more significant increase in roughness can be observed, but also a greater separation between the columns and the number of particulates on the surface becomes noticeable [46,47]. This result shows a more arranged topology when the thin film is deposited on a substrate with 10 min of UV treatment. Figure 4b presents a topography with an irregular pattern produced by the roughness of the substrate, leading to the formation of agglomerates on the surface of the film. A more homogeneous growth was presented since the surface is observed to be smooth without microdefects such as fractures or the growth of macroparticles.

The homogeneous topology growth presented in Figure 4a is related to a shrinkage phenomenon of the polymeric substrates due to the temperature changes produced via the interaction with the PVD-RMS plasma environment [24,28,29,38]. The most common post-cured treatments reported for photosensitive polymers are UV and heat treatments [21,22,28,40,42,48,49]. Different studies [22,29] have shown that any post-cure treatment is effective in the obtention of a polymer with improved final properties compared to a green-state polymer structure. However, the interaction between the green-state polymer and any post-cure treatment results in an undesired effect known as shrinkage [24,29,48].

The shrinkage phenomenon is inherent to the curing of methacrylate-based photopolymers [18,50]. Due to the polymerization process, the start liquid resin is converted into a solid, resulting in a density change that reduces the overall volume, producing volumetric shrinkage upon curing and drying [39,50]. These phenomena lead to internal stresses and distort consequences [24,29,38,48]. Afterward, with a post-cured process, a second polymerization process is generated throughout the entire structure of the as-printed part, particularly in the areas that have different curing portions, as shown in Figure 1, increasing the distortion and internal stresses produced via the first polymerization, reducing the surface finish, and magnifying the surface roughness values of the final part [18,20,28,38,42].

Compared to UV post-cured treatments [21,22,28,40,42,48,49], studies have established that particularly thermal post-curing treatments reduce heterogeneity and anisotropy, inducing undesired higher shrinkage strains and representing an accelerated aging mechanism [51], consequently, increasing the shrinkage and the risk of cracking [52]. The shrinkage of successively built layers produces residual stresses that accumulate, generating strain deformations leading to a considerable curl-distortion of the multi-layered polymer parts [41,53].

One of the main characteristics of PVD-RMS plasma is the generation of heat; the ionized gas realizes high temperatures via the high-velocity movement of free electrons, the interaction of sputtered particles with the gas results in gas heating, and the constant bombardment of the sputtered atoms during the deposition process is released in form of heat in the substrate [7,8,9,25,43]. Therefore, the result of the homogeneous topology was not only the result of this accumulation of stresses, but also the result of the continuous bombardment of the molecules that form the thin film, added to the working temperature of the reactive PVD-RMS plasma, and the way the film mimics the topography of the substrate [28,41,42,51,54].

Figure 4c,d presents the EDS spectrum corresponding to the samples analyzed, showing no presence of the substrate elements with the characteristic peaks attributed to the thin film, Zn and O. The effect of the PVD-RMS plasma on the thicknesses of the Zn/ZnO samples was also analyzed via SEM. Figure 4e,f shows the SEM BSE images of the thicknesses obtained in Zn/ZnO thin films. Thicknesses between ~1.3 µm and ~580 nm were obtained for the Zn/ZnO films when they were deposited on substrates with and without UV treatment, respectively. A greater thickness was obtained when the thin films were deposited on photosensitive polymeric substrates with 10 min of UV treatment, as seen in Figure 4e. A study carried out by R. Castro et al. [55] suggests that the phenomenon of higher thicknesses is related to a higher surface roughness of the substrate.

A study performed by Zhao J. et al. [41] showed that photopolymer post-cured parts tend to have higher surface roughness values compared to green-state parts due to the shrinkage phenomena [18,24,29,48,50]. Although Zn and ZnO materials are deposited via PVD-RMS under the same conditions on both substrates; differences in roughness caused via the shrinkage phenomena and post-cure treatments resulted in the sputtered atoms being deposited under different growth dynamics caused by greater availability of the substrate area presented for deposition due to the increase in roughness. The results obtained by R. Castro et al. [55] reveal a possible explanation for the phenomenon presented in the thickness differences obtained for Zn/ZnO thin films, related to an increase in roughness caused via the shrinkage phenomena accumulation of photosensitive polymeric substrates with 10 min of UV treatment.

To analyze the effect of the UV treatment on the microstructural properties of the Zn/ZnO thin films and the changes produced via the PVD-RMS plasma, an uncoated photosensitive polymeric substrate was used to examine the differences found in the rough surface in comparison with the Zn/ZnO thin films using 3D AFM. Additionally, the coated samples were analyzed to compare growth similarities, according to Thornton’s diagram, and to observe if PVD-RMS plasma treatment modifies the growth of the thin films deposited on polymeric substrates with different UV treatments [56]. Figure 5 presents the 3D topology images obtained via AFM, together with the surface roughness parameters (in values of ‘Rq’ = root mean square roughness), made on a scan area of 400 μm^2^. Approximate values of surface roughness (Rq) of 0.0241, 0.0956, and 0.205 µm were obtained for the naked substrate (a) and the thin films ((b) and (c)), respectively.

The smoothest surface corresponds to the uncoated substrate in Figure 5a. Subsequently, due to the growth nature of the films, the surface roughness of the samples shown in Figure 5b,c increases as a result of the formation of ZnO columns with irregular boundaries and amorphous structure, related to zones 1a/1b, based on the thin film growth model presented by Thornton, characterized as being developed at low Ar pressures and low temperatures [46,47,56,57]. The same growth characteristics were observed in the Zn/ZnO films produced, showing that the substrate’s post-curing treatment does not affect the type of growth of the thin films. Nevertheless, the thin film (c) produced over a polymeric substrate with UV treatment results in higher values of the surface roughness of the samples.

The increased roughness of sample (c) was related to the surface deformation caused by the shrinkage stress accumulated during the polymer substrate manufacture and post-processing. The work by D. Karalekas et al. [42] explains this correlation since their research demonstrates that residual stresses generated during polymer building and post-curing are responsible for superficial creep distortions, particularly those related to thermal post-curing treatments. In this way, it can also be considered that the plasma of the PVD-RMS technique has caused surface deformation, during the deposition of thin films, due to the plasma temperatures [21,25,29,30]. The accumulation of stresses produced by the UV treatment of sample (c) caused a surface deformation in the polymer, which due to growth phenomena, allowed the films to adhere and adapt to the substrate surface to grow with columns of different sizes that resulted in an increase in surface roughness [12,13,14]. Although the change in roughness was significant via the post-cured treatment of the substrates, the interaction between the plasma environment and the substrates did not produce a difference in the observed growth of the thin films.

### 3.2. Effect of UV Treatment on the Adhesion Properties of Zn/ZnO Thin Films

Rockwell marks were observed using SEM-EDS for qualitative analysis and to identify how the effect of the PVD-RMS plasma modifies the adhesion properties of the samples. Figure 6 shows the SEM-BSE images of the Zn/ZnO thin films deposited on two photosensitive polymer substrates with different UV treatments.

Coatings exhibited acceptable failures according to the VDI 3198 standard, with the presence of cracks and delamination around the indentation marks. The same phenomenon was identified for both samples, as observed in the images obtained via SEM at low magnification. However, the degree of delamination increased slightly when the thin film was deposited on a photosensitive polymer substrate with 10 min of UV treatment.

The adhesion properties of thin films can be affected by different factors [45,58]. In the research carried out by Thouless et al. [52], it is mentioned that the adhesion of a thin film will depend on the residual stress accumulated during its production, which, in turn, is related to various deposition parameters, such as growth temperature, the temperature of the substrate, deposition rate, etc. [45]. It has been shown that those thin films that grow densely, uniformly, and with limited thickness dimensions tend to reduce the distribution of stresses during their production, thus presenting better adhesion properties [45,58,59].

However, in addition to the deposition parameters, the substrate is also a determining factor in this research. The interaction between PVD-RMS plasma with a photosensitive polymer substrate causes the substrate itself to accumulate its own stresses, affecting the interfacial bonds between substrate and film and, therefore, the final adhesion properties [59]. The origin of residual stresses is related to a discrepancy in the coefficient of thermal expansion between substrate and film. Its magnitude and size depend as well on the thickness and size of the substrate and coating [10,60,61,62].

Since the plasma performs a thermal post-curing treatment, its effect on the polymeric substrate reduces the superficial heterogeneity and anisotropy, inducing strains [51] and, consequently, increasing the shrinkage [52], producing residual stresses that accumulate, generating strain deformations of the multi-layered polymer parts that then were transferred to the films during the deposition. Differences in roughness caused by the shrinkage also resulted in a greater thickness obtained for sample (a), leading to high residual film stresses generated during deposition [45,58]. In addition, the columnar morphology presented with high roughness in sample (a) is generally not desirable. Local stresses can be found in films with non-homogeneous growth, allowing poor adhesion [45].

The 10 min of UV treatment implemented in only one polymeric substrate (sample a) causes them to behave as two different substrates. A study [59] has shown that two thin films of the same material, deposited under the same growth parameters, can present completely opposite adhesion properties when deposited on different substrates. Although two polymeric substrates of the same material were used in this investigation, the other thermal treatments used during their production resulted in a series of changes in the microstructural and adhesion properties of the thin films.

The image magnification was increased to carry out a deeper analysis in a specific zone of the indentation footprint [red square in Figure 6 using the elemental mapping method via EDS equipped in SEM to identify the elements present in the microfractures and delamination. Elemental mapping shows the presence of carbon in the fracture zones and a decrease in the elements zinc and oxygen. Since the element carbon comes from the substrate, the detachment of the film and the adhesion layer was confirmed. These results have demonstrated that the changes in curing treatment parameters of the sample in Figure 6a produced a fracturing effect in the adhesion tests, showing that the adhesion properties of the films decrease when photosensitive polymers with 10 min of UV treatment are used as substrates since more catastrophic failures were found, presenting extended delamination and fractures at the vicinity of the imprint, indicating a poor interfacial adhesion between the film and the substrate.

The evaluation criteria of the Rockwell test, following the VDI-3198 standard, says that the deposited Zn/ZnO coatings presented in Figure 6 belong to the category of acceptable failures; the sample in Figure 6b showed stronger interfacial bonds with fewer cracks and delamination and, therefore, a higher degree of adhesion [59,63].

The study by Vidakis et al. [59] could adequately explain the differences observed between the two polymeric substrates used with different curing treatments. Throughout this research, it has been observed that PVD-RMS plasma modifies the surface properties of a photosensitive polymeric material used as a substrate. So, because of the final thin film properties, such as increased roughness, thickness and weakening of the structure of the film, a substrate with 10 min of UV treatment is not favorable for the adhesion properties of the thin film [46].

The degree of adhesion of the films is not related to a single microstructural property but a combination of them. Therefore, the sum of the properties found on the sample (a) (greater thickness, greater roughness, and less homogeneous surface with irregular growth and fractures of the substrate) has caused the decrement in the adhesion properties of the film when it is deposited on a substrate with 10 min of UV and plasma treatment.

One fact to highlight is that, with fewer defects, the accumulation of residual stresses in the films decreases, and this promotes better adhesion. If each of the indicated microdefects has caused residual stresses not only in the substrate but also in the film, it is normal to find that sample (a) has presented the lowest degree of adhesion [38,63,64].

## 4. Conclusions

The effect of the plasma produced via the PVD-RMS technique as an alternative post-curing treatment on photosensitive polymeric substrates was demonstrated. Changes in Zn/ZnO thin films varied according to the post-cured treatments of the substrates. Superficial fractures were observed in Zn/ZnO thin films deposited on a polymeric substrate with 10 min of UV treatment. The phenomenon was related to a deterioration of the substrate properties, as a consequence of an abrupt change in temperature by the interaction with the PVD-RMS plasma environment, inducing higher shrinkage strains and cracking. The plasma treatment showed an effect on the topology changes of the films. A particular homogeneous topology pattern was observed in the sample with 10 min of UV treatment. The singular topology was not only the result of residual stresses accumulated on the successively built layers but also the way the film mimics the topology of the substrate. Thicknesses between 1.3 µm and 580 nm were obtained for the Zn/ZnO films when they were deposited on substrates with and without UV treatment, respectively. The thickness differences were related to an increase in roughness caused by the shrinkage phenomena of photosensitive polymeric substrates with 10 min of UV treatment. The plasma treatment showed an effect on the roughness changes of the films. Surface roughness (Rq) of 0.205 and 0.0956 µm were obtained for the thin films deposited on substrates with and without UV treatment, respectively. The accumulation of stresses produced via the UV treatment caused a surface deformation in the polymer, allowing the films to grow with columns of different sizes that resulted in an increase in surface roughness. The columnar growth of the thin films was not altered from the plasma environment. The adhesion tests showed detachment and microfractures in the analyzed samples, thus obtaining acceptable failures according to the VDI 3198 standard for all samples. The plasma effect showed a slight decrease in the adhesion properties of the films. The sputtering technology allows a high degree of freedom regarding the deposition materials and substrates employed [65]. With the suitable changes, an alternative methodology can be applied to allow the use of polymeric substrates obtained via stereolithography, without altering the general properties of the substrate, for the deposition of thin films via the sputtering technique and its use in different applications that require the direct creation of functional prototypes [66]. Since the thin films produced present good properties in general and accepted adhesion failures, we expect that they could be suitable for applications that require high surface roughness.

## Figures and Tables

**Figure 1 polymers-15-02283-f001:**
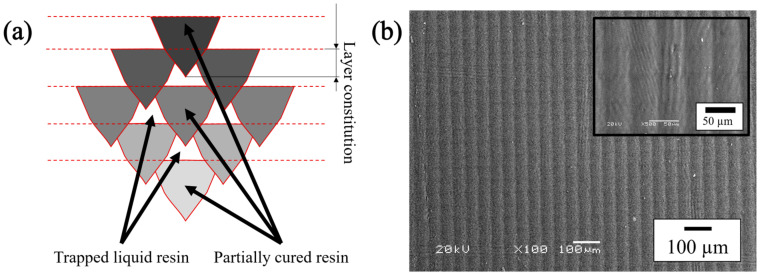
The final properties of the polymer substrate change between the surface and along the 3D structure. (**a**) Polymer showing the difference polymerized portions throughout the pattern; (**b**) Surface patterns from a polymer sample seen by backscattered electron imaging on scanning electron microscopy (SEM-BSE) with low and high magnification.

**Figure 2 polymers-15-02283-f002:**
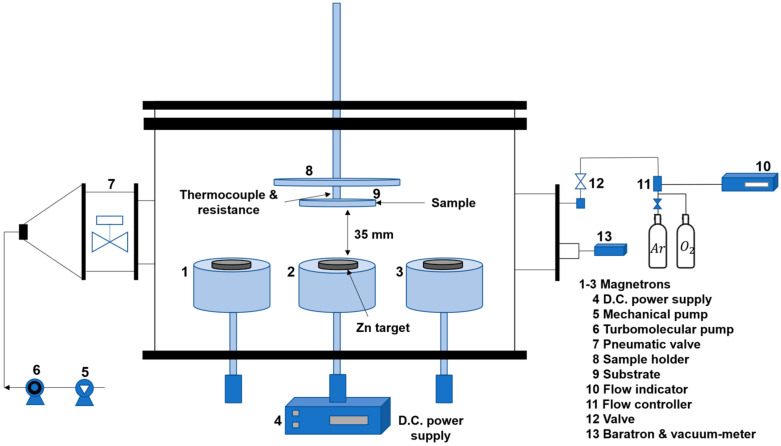
Diagram of the deposition setup and its main components.

**Figure 3 polymers-15-02283-f003:**
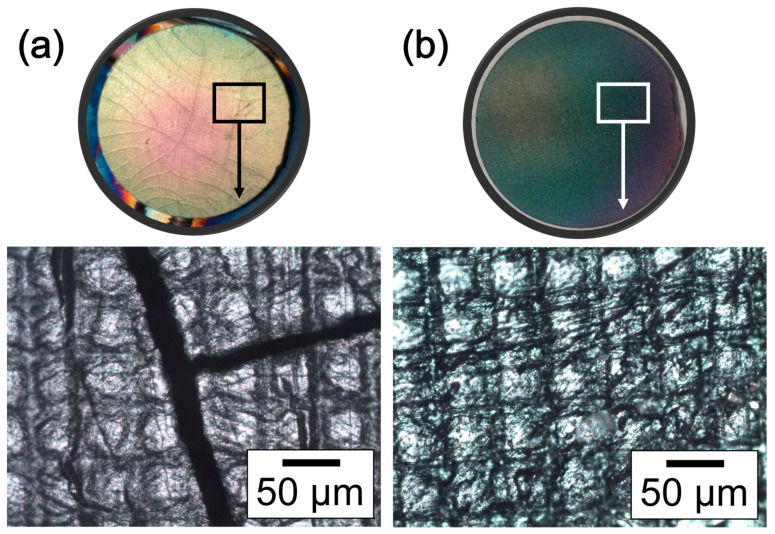
Macroscopic view and OM images show differences on the sample’s surfaces. Polymer substrate with (**a**) 10 min of UV treatment and (**b**) without UV treatment.

**Figure 4 polymers-15-02283-f004:**
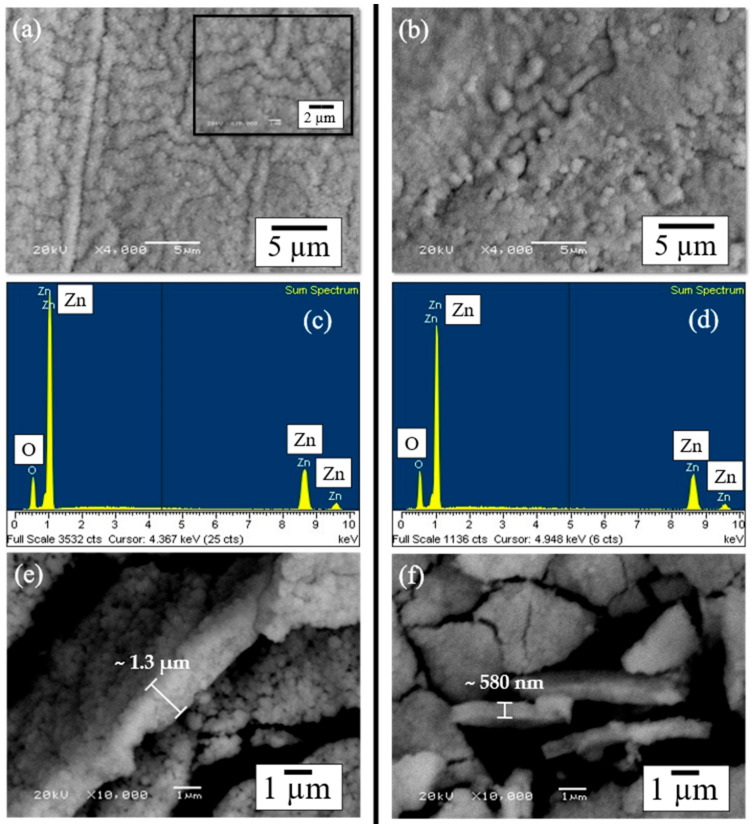
SEM-BSE surface views at high magnification showing differences in the film’s surface topology. (**a**) Polymer with 10 min of UV treatment exhibits a homogeneous growth, and (**b**) polymer without UV treatment presents a topology with an irregular pattern. Elemental mapping via EDS in the area. (**c**) Polymer with 10 min of UV treatment and (**d**) polymer without UV treatment. SEM micrographs showing the thickness of the samples. (**e**) Polymer with 10 min of UV treatment presents a ~1.3 µm thickness, and (**f**) polymer without UV treatment presents a ~580 nm thickness.

**Figure 5 polymers-15-02283-f005:**
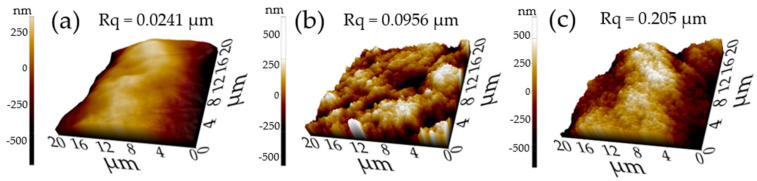
Three-dimensional AFM images (20 μm × 20 μm) of (**a**) uncoated polymer substrate, (**b**) Zn/ZnO thin film on polymer substrate without UV treatment, and (**c**) Zn/ZnO thin film on a polymer substrate with 10 min of UV treatment in order to compare the changes of the roughness parameters between samples.

**Figure 6 polymers-15-02283-f006:**
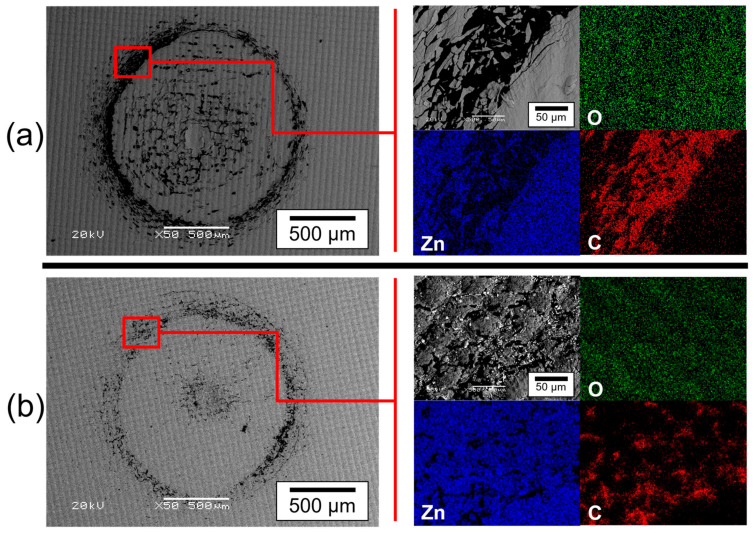
SEM-BSE images of the adhesion test of the Zn/ZnO thin films at low magnification, using a Rockwell durometer following the VDI 3198 standard. (**a**) Polymeric substrate with 10 min of UV treatment and (**b**) polymeric substrate without UV treatment.

**Table 1 polymers-15-02283-t001:** Printer parameters used for polymer substrate production.

Production Parameter	Units
Technology	LED Display Photocuring
Light source	Integrated UV light (405 nm)
XY axis resolution	0.0047 mm (2560 × 1440 px)
Z axis accuracy	0.00125 mm
Thickness per layer	0.05 mm
Exhibition time	8 s
Lower exposure time	60 s
Print speed	22.5 mm/h
Total printing time per lot	~30 min

## Data Availability

The data presented in this study are available on request from the corresponding author.

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
