# Peer review of "Sputtering Plasma Effect on Zinc Oxide Thin Films Produced on Photopolymer Substrates"

_polymers, 2023, doi:10.3390/polym15102283_

Round 1

Reviewer 1 Report

 This work presents a post-cured treatment alternative for photopolymer substrates considering the plasma produced by the sputtering process. The sputtering plasma effect was discussed by analyzing the properties of zinc/zinc oxide (Zn/ZnO) thin films deposited on photopolymer substrates, with and without ultraviolet (UV) treatment as a post-treatment process, after manufacturing. The results showed the effect of plasma as a post-cured treatment alternative: fractures were found in thin films deposited on polymers with previous UV treatment. In the same way, the films showed a repetitive printing pattern, due to the phenomenon of polymer shrinkage caused by the sputtering plasma. There are some comments toward the research.

1.     Please compare this work with recent related work to further clarify the innovation and performance advantages.

2.     In SEM, Figure 4(a) shows a homogeneous growth and a more arranged topology when the thin film is deposited on a substrate with  10 min of UV treatment. While, in 3D-AFM Figure 5 (c) the UV treatment sample shows a higher roughness (Rq=0.205) than the without UV treatment sample. Are the results contradictory?if not, please explain it.

3.     In Figure 5, the 3D-AFM picture, The font in the picture is not clear, please revise it.

4.     In Figure 4 (e) and (f), it is suggested to mark to indicate the thickness of 1.3 um and 580 nm, respectively. 

5.     For the EDS results, what the difference between Figure 4 (c) and 4(d).

Author Response

Response to Reviews Manuscript polymers-2384736  Sputtering Plasma Effect on Zinc Oxide Thin Films Produced on Photopolymer Substrates, Juan Jesús Rocha-Cuervo, Esmeralda Uribe-Lam *, Cecilia D. Treviño-Quintanilla, Dulce Viridiana Melo-Maximo

 Dear Reviewer, #1:

We thank the Reviewers for their time and effort assesing our manuscript. We greatly appreciate their feedback—it decisively helped us to significantly improve our manuscript. We addressed the best we could all comments and requests for clarification and improvement. Line-by-line responses are furnished below, and recaps of the changes introduced for clarity. We hope the current version of our manuscript is acceptable for publication.

Reviewer #1:

This work presents a post-cured treatment alternative for photopolymer substrates considering the plasma produced by the sputtering process. The sputtering plasma effect was discussed by analyzing the properties of zinc/zinc oxide (Zn/ZnO) thin films deposited on photopolymer substrates, with and without ultraviolet (UV) treatment as a post-treatment process, after manufacturing. The results showed the effect of plasma as a post-cured treatment alternative: fractures were found in thin films deposited on polymers with previous UV treatment. In the same way, the films showed a repetitive printing pattern, due to the phenomenon of polymer shrinkage caused by the sputtering plasma. There are some comments toward the research.

  1. Please compare this work with recent related work to further clarify the innovation and performance advantages.

R:\ Thank you for your feedback. We carefully considered your suggestion; the following paragraph is included in the manuscript in the conclusions section that highlights a comparison between related and published investigations to clarify the innovation of the phenomenon described by our research. In more detail:

“The sputtering technology allows a high degree of freedom regarding the deposition materials and substrates employed [65]. With the suitable changes, an alternative methodology can be applied to allow the use of polymeric substrates obtained by stereolithography, without altering the general properties of the substrate, for the deposition of thin films by the sputtering technique and its use in different applications that require the direct creation of functional prototypes [66]. Since the thin films produced present good properties in general and accepted adhesion failures, we expect that they could be suitable for applications that require high surface roughness.”

  1. [Fabrication of Highly Porous and Pure Zinc Oxide Films Using Modified DC Magnetron Sputtering and Post-Oxidation].
  2. [Additive Manufacturing Technologies – Rapid Prototyping to Direct Digital Manufacturing].

 Summary of changes made to the manuscript: text and new Ref. [65] and Ref. [66] was added to support the  comparison with related research and to clarify the innovation and performance advantages of our results.(Pages 12, lines 431-439 of the amended manuscript).

2.- In SEM, Figure 4(a) shows a homogeneous growth and a more arranged topology when the thin film is deposited on a substrate with 10 min of UV treatment. While, in 3D-AFM Figure 5 (c) the UV treatment sample shows a higher roughness (Rq=0.205) than the without UV treatment sample. Are the results contradictory?if not, please explain it.

R:\ Thank you for your feedback, authors appreciate your input, answering at your question, we consider that the behavior is not contradictory. The explanation in detail is related to the excess of energy present in the samples that were treated by UV and later subjected to plasma.The increased roughness of the sample was related to the effect of two phenomena. First, the polymer surface deformation caused by the shrinkage stress accumulated during the UV treatment post-processing, and secondly, the growth nature of the films. The Zn/ZnO thin film produced over a polymeric substrate with UV treatment results in the formation of columns with irregular boundaries and amorphous structure, causing the repetitive (homogeneous) surface growth, in form of labyrinth and with greater grain limits, with increased surface roughness in comparison with the polymeric sample without UV treatment. With these results we wanted to explain that, although excess curing allows us to better organize the superficial structure of the films, the greater organization generates also greater surface roughness in general. No changes were made in the document if it necessary to add an extra explanation, please let us know.

Summary of changes made to the manuscript: no changes were made in the manuscript.

3.-In Figure 5, the 3D-AFM picture, the font in the picture is not clear, please revise it.

R:\ Thank you for your feedback, we agree with the Reviewer that the font in the picture was not clear. To make the appropriate changes to clarify the image, its font was modified, the boxes in the initial letters or markers were eliminated and the roughness that we want to highlight in each image was placed through text. The X and Y scales of the images were also placed with a higher font for easier reading and the Z vertical roughness scale was increased in size for easier interpretation. We hope that these modifications are sufficient to allow the correct visualization and compression of our image. New figure 5 in detail: Please see the attachment.

Summary of changes made to the manuscript: the figure 5 in page 9 line 286-291 of the manuscript was replaced with figure with the after mentioned modifications.

4.- In Figure 4 (e) and (f), it is suggested to mark to indicate the thickness of 1.3 um and 580 nm, respectively. 

      R:\ Thank you for your feedback, these are excellent suggestions to clarify our images. Modifications were made in figure 5, images (e) and (f), the boxes that mention the thicknesses of the film in the upper part were eliminated and measurement dimensions were included in the measured area of ​​each film, including the measurement in text with units next to it, which corresponds to each film thickness. We hope that these modifications are sufficient to allow the correct visualization and compression of our image. New figure 4 in detail, section (e) and (f): Please see the attachment.

Summary of changes made to the manuscript: the figure 4 (e) and (f) in page 7 line 202-208 of the manuscript was replaced with figure with the after mentioned modifications.

5.-For the EDS results, what the difference between Figure 4 (c) and 4(d).

      R:\ Thank you for your feedback. The answer to your question is mentioned in detail below: in figure 4 section (c) and (d) we try to present the composition analysis of each one of the films is shown, where the presence of the oxygen and zinc signals demonstrates the existence of zinc oxides in each film analyzed. For the EDS results, the Zn and O labels were moved sideways to clearly exhibit the signal differences between Figure 4 (c) and 4 (d). The oxygen and zinc signal varies from each film, the zinc signal is higher in the film that does not present UV treatment, which indicates a greater presence of pure zinc in this film instead of zinc oxide, while in the film that has UV and plasma treatment the zinc signal is lower, and the oxygen signal is higher, so there is a greater amount of zinc oxide. Finally, we want to illustrate how element signals of Zn and O slightly changed when the thin films are deposited in samples without UV treatment, compared to the sample produced with UV treatment. New figure 4 in detail, section (c) and (d): Please see the attachment

Summary of changes made to the manuscript: the figure 4(c) and (d) in page 7 line 202-208 of the manuscript was replaced with figure with the after mentioned modifications.

Reviewer 2 Report

The paper is devoted to the study of the sputtering plasma effect on Zinc Oxide thin films. This subject is important for the creation of various Zn/ZnO nanostructures on the surface of polymers. It has carried out a lot of experimental work and compiled a good review of the literature.

The results obtained are not in doubt and are clearly presented. The paper can be published in its current form.

Author Response

Response to Reviews Manuscript polymers-2384736  Sputtering Plasma Effect on Zinc Oxide Thin Films Produced on Photopolymer Substrates, Juan Jesús Rocha-Cuervo, Esmeralda Uribe-Lam *, Cecilia D. Treviño-Quintanilla, Dulce Viridiana Melo-Maximo

Dear Reviewer #2

We thank the Reviewers for their time and effort assesing our manuscript. We greatly appreciate their feedback.

Reviewer #2: 

The paper is devoted to the study of the sputtering plasma effect on Zinc Oxide thin films. This subject is important for the creation of various Zn/ZnO nanostructures on the surface of polymers. It has carried out a lot of experimental work and compiled a good review of the literature.

The results obtained are not in doubt and are clearly presented. The paper can be published in its current form.

R:\ Thank you for your feedback, all the authors appreciate Reviewer’s comments, we also value the time dedicated to reading, analyzing and understanding our scientific article.